# Menin–MLL1 Interaction Small Molecule Inhibitors: A Potential Therapeutic Strategy for Leukemia and Cancers

**DOI:** 10.3390/molecules28073026

**Published:** 2023-03-28

**Authors:** Qing Shi, Meiqi Xu, Zhijian Kang, Manjie Zhang, Yakun Luo

**Affiliations:** 1National Health Commission Key Laboratory of Molecular Probes and Targeted Diagnosis and Therapy, Harbin Medical University, Harbin 150001, China; 2Department of Urology, The Fourth Affiliated Hospital of Harbin Medical University, Harbin 150001, China; 3College of Pharmacy, Harbin Medical University, Harbin 150080, China

**Keywords:** molecule inhibitor, menin-MLL1 complex, leukemia, cancer, therapeutics strategies

## Abstract

Encoded by the *MEN1* gene, menin protein is a fusion protein that is essential for the oncogenic transformation of mixed-lineage leukemia (MLL) and leads to acute leukemia (AL). Therefore, accumulating evidence has demonstrated that inhibition of the high-affinity relationship between menin and mixed-lineage leukemia 1 (MLL1 and KMT2A) is an effective treatment for MLL-rearranged (MLL-r) leukemia in vitro and in vivo. Meanwhile, recent studies found that menin–MLL1 interaction inhibitors exhibited a firm tumor suppressive ability in specific cancer cells, such as prostate cancer, breast cancer, liver cancer, and lung cancer. Overall, it seems to serve as a novel therapeutic means for cancers. Herein, we review the recent progress in exploring the inhibitors of small molecule menin–MLL1 interactions. The molecular mechanisms of these inhibitors’ functions and their application prospects in the treatment of AL and cancers are explored.

## 1. Introduction

The *MEN1* gene is one of the key pathogenic genes of multiple endocrine neoplasia 1 (MEN1) syndromes [1]. In 1988, the *MEN1* gene was found in chromosome 11q13 [2], and it was positionally cloned in 1997 [3,4]. With a span of 9.8 kb and 10 exons, the position of the *MEN1* gene is at chromosome 11q13. Exon 1 and the distal part of exon 10 are non-coding sequences. It encodes the menin protein. The *MEN1* gene has a wide expression profile and a tissue-specific menin function, as proven in biochemical and genetic studies [5].

Menin is found to be an essential influencer in the pathogenesis of leukemia [6]. It has been demonstrated that menin is essential in the oncogenic transformation of fusion proteins in mixed-lineage leukemia (MLL), including MLL-ENL, MLL-AF4, MLL-AF6, and MLL-AF9, subsequently leading to acute leukemia (AL). In the clinical experience of acute lymphoblastic leukemia (ALL) and acute myeloid leukemia (AML), the *MLL* gene’s rearrangement is commonly observed in chromosome translocation [7,8], with acute myeloid leukemia (AML) incidence of approximately 10% in adults and 70% in infants [9,10]. There is an unfavorable prognosis for leukemia patients suffering from *MLL* translocation, and they have poor responses to available therapeutic means. According to biochemical research, menin is related to histone methyltransferases (HNT) MLL1 and MLL2 in the trithorax family. Furthermore, the crystal structure analysis showed that it could perform a direct binding with the N-terminus of MLL1 in its fusion proteins and is critical in recruiting them with target genes, such as *MEIS1* and *HOXA9* [11]. Therefore, targeting the interaction between menin and MLL1 is considered an effective method for rearranging MLL in treating acute leukemias. During the past decades, researchers made huge efforts to develop small molecule inhibitors such as MI-2, MI-3, MI-525, or MI-503, which directly bind to menin and inhibit the interaction between menin and MLL1. Therefore, targeting the interaction between menin and MLL1 is considered an effective method for MLL-r leukemia. In recent decades, researchers have developed small molecular inhibitors, such as MI-2, MI-3, MI-525, or MI-503, which perform a direct binding with and limit the interaction between menin and MLL in human cells and in vivo.

Meanwhile, more and more studies have demonstrated that menin participates in the carcinogenesis of solid tumors. As a tumor suppressor, menin is capable of direct control of cell development in the pituitary gland, islet, thyroid gland, and other endocrine organs. Moreover, it has been reported that menin promotes tumorigenesis in specific contexts, including cancers in the prostate, breast, and liver. The currently available data show that menin is crucial in many aspects of the carcinogenesis of different cells, but its functioning mechanism has not been fully elucidated. As an auxiliary factor of the histone methyltransferase of MLL1 and MLL2, the inhibitory effect of menin–MLL1 should be further studied, since many diseases, including cancer, are closely related to the activities of MLL1 and MLL2. Overall, menin–MLL1 inhibitors are believed to have more comprehensive clinical benefits. Subsequently, in the research or treatment of prostate, breast, and liver cancers, menin–MLL1 inhibitors have made some progress until now. 

It is necessary to review the recently developed and biological mechanism progress of menin–MLL1 interaction molecular inhibitors, which might offer possibilities for the more in-depth development of menin–MLL1 interaction-based molecular inhibitors in treating cancer and MLL-r leukemia.

## 2. Type of Hydroxymethyl and Aminomethyl Piperidine Inhibitors

The hydroxymethyl and aminomethyl piperidine compound is the first generation of the small molecule inhibitors of menin–MLL1 interactions. Borkin et al. obtained MIV-1 and MIV-2 active compounds via high-throughput screening from 288,000 small molecule compounds. After subsequent structural optimization, a series of hydroxymethyl and aminomethyl piperidine compounds were obtained [12,13]. Herein, we will enumerate some inhibitors and how they can be used in treating cancers, including leukemia. The structures of hydroxymethyl and aminomethyl piperidine inhibitors are shown in Table 1.

### 2.1. M-525

The M-525 is highly selective to leukemia cells with MLL1 fusion protein (Table 1). The IC50 values of M-525 in inhibiting the leukemia cell lines (MV-4-11) carrying MLL fusion reach 3 nM. M-525 could effectively inhibit the expression of *HOX* and *MEIS1* genes in the MOLM-13 cell that carries MLL-AF9 fusion and the MV-4-11 cell that has MLL-AF4 fusion [14]. The transcriptional expression of *MEIS1* and *HOX* genes was crucial for the development of leukemia (Figure 1).

### 2.2. M-808

M-808 produces further optimization on the structural basis of M-525 (Table 1). Therefore, in menin–MLL1 interactions, M-808 performs in a similar way to the cocrystal structure of M-525. The IC50 values of M-808 in inhibiting the leukemia cell lines (MOLM-13 and MV-4-11) carrying MLL fusion reach 1 and 4 nM, respectively. In addition, in the xenograft model of MV-4-11 leukemia in mice, certain tumors are successfully limited [15] (Figure 1).

### 2.3. M-89

M-89 is a noncovalently bound small molecule inhibitor of menin–MLL1 based on previous structures designed and synthesized (Table 1). M-89 potently inhibited the development of MV-4-11 and MOLM-13 cells, with IC50 values of 25 nM and 54 nM. In contrast, the IC50 value of M-89 in HL-60 cells is only 10.2 μM. Thus, M-89 showed strong cell growth inhibition activity in MV-4-11 and MOLM-13 leukemia cell lines carrying the MLL fusion protein. M-89 can also block the development of leukemia by inhibiting the expression of *MEIS1* and *HOX* genes [16,17] (Figure 1).

### 2.4. M-1121

As a covalent oral active inhibitor of the interaction between menin and MLL1, M-1121 can completely regress a tumor in the long term (Table 1). The IC50 values of M-1211 in inhibiting the leukemia cell lines (MOLM-13 and MV-4-11) carrying MLL fusion reach 51.5 and 10.3 nM, respectively. To determine its selectivity and cell activity, researchers measure the antiproliferation activity of M-1121 in three kinds of wild-type MLL cell lines and four kinds of MLL-r. The results showed that M-1121 could effectively limit the growth of MLL-r leukemia cells but have no inhibitory effect on MLL wild-type cells in 10 μM. This indicates the selectivity of M-1121 [18] (Figure 1).

## 3. Type of Thiophenpyrimidine Inhibitors

Borkin et al. discovered a small molecule inhibitor at the interface of the interaction between menin and MLL1 [19]. The compound MI-1 (IC50 = 1.9 μM) was screened by high-throughput screening from 48,000 compounds, and it was the first generation of thiophenpyrimidine small molecule inhibitors [20]. This compound is a new type of menin small molecule inhibitor, with good oral bioactivity and strong target binding activity, which can be called the second generation of menin small molecule inhibitors [17]. The structures for thiophenpyrimidine inhibitors, which we selected to review, are shown in Table 2.

### 3.1. MI-2/MI-3

In human acute leukemia (AML) cell lines with different MLL translocations, both MI-2 and MI-3 exhibit effective and dose-dependent growth inhibition [20,21,38,39]. The IC50 values of MI-2/MI-3 in inhibiting the leukemia cell lines (MV-4-11) carrying MLL fusion reach 446 or 648 nM. The expression of MEIS1 and HOXA9 can be significantly reduced by treated THP-1 cells. THP-1 is a mononuclear cell line in human peripheral blood, which originated from patients suffering from acute monocytic leukemia. However, REH, HAL-01, ME-1, Kasumi-1, and other human acute leukemia cell lines that lack MLL fusion and low HOXA9 expression are less affected by MI-2 and MI-3 [20] (Figure 2A). In addition, synergistic cytotoxic activity is observed during the simultaneous inhibition of histone deacetylase (HDAC) and menin–MLL in the interaction process, especially for AML cells that carry MLL rearrangements. In MLL-rearranged AML cells, there is a high degree of synergy between MI-3 and the HDAC inhibitor chidamide inhibiting the menin–MLL interaction [40].

Besides the above-mentioned antileukemia activity, MI-2 and MI-3 also displayed antitumor abilities in specific contexts. Studies demonstrated that the combination of MI-2 and the inhibitors (ICG001) of β-catenin restricted mice salivary gland tumor cell spheroids from generating and inhibited the growth of tumor cells in the human head and neck in a concentration-dependent manner [20,22] (Figure 2B). Furthermore, it has been reported that MI-3 specifically inhibits the proliferation of the Kras mutant lung cancer cells but has little impact on inhibiting the proliferation of wild-type Kras lung cancer. Mechanistically, histone H3 lysine 4 trimethylation (H3K4me3) is reduced due to MLL deficiency, thus inhibiting the expression of Ras protein-specific guanine nucleotide-releasing factor 1 (Rasgrf 1). Rasgrf 1 plays a crucial role in the activation of *Kras* downstream pathways and their cancer-promoting activity [23] (Figure 2C). The structures of MI-2 and MI-3 are shown in Table 2.

### 3.2. MI-136

The abnormally activated androgen receptor (AR) activity is the main driving factor of resistant prostate cancer (CRPC). As a co-activating factor of AR signals, the AR performs a direct interaction with the MLL complex via the menin–MLL subunit. The expression of menin in hormone-naive prostate cancer and benign prostate tissue is lower than that in castration-resistant prostate cancer. The high menin expression is associated with a poor survival rate in prostate cancer patients. MI-136 treatment blocked the AR signaling pathway of prostate cancer cell lines (LNCaP and VCaP) in mice [5] and restricted the proliferation of castration-resistant tumors [5] (Figure 2D).

Studies have shown that MI-136 can also inhibit endometrial tumor growth in vitro and in orthotopic models in vivo. The IC50 of MI-136 on these endometrial cancer organoids was 4.5 μM. In the mechanism, MI-136, the inhibitor of menin–MLL, directly downregulated the expression of multiple components of the HIF pathway, such as Nos2/Nos3/Cav1, thereby inhibiting tumor growth [24] (Figure 2E). The structure of MI-136 is shown in Table 2.

### 3.3. MI-538

The activity and selectivity of MI-538 were significantly higher than those of MI-136 and showed significant effects in the MLL leukemia mouse model. The cell proliferation of human MLL leukemia cell lines (MV-4-11 and MOLM13) can be inhibited by MI-538, with two various MLL translocations (MLL-AF4 and MLL-AF9, respectively). The IC50 values of MI-538 in inhibiting the leukemia cell lines (MV-4-11) carrying MLL fusion reached 21 nM. Treatment with MI-538 led to a strong downregulated expression of *MEIS1* and *HOXA9* genes (Figure 2F). Meanwhile, MI-538 significantly reduced MV-4-11 tumor volume in vivo and did not cause obvious toxic symptoms, which significantly inhibited the progression of leukemia. The MI-538 is a promising treatment for MLL-r leukemia [25]. The structure of MI-538 is shown in Table 2.

### 3.4. MI-463/MI-503

It is reported that MI-463 and MI-503 can inhibit the menin–MLL interaction, thus blocking tumor growth by MV-4-11 cells in mouse xenograft models and improving the survival rate. The IC50 values of MI-463/503 in inhibiting the leukemia cell lines carrying MLL fusion reached 15.3 or 14.7 nM. Mechanically, due to the high affinity of MI-503 and MI-463 relative to menin, the menin–MLL complex interaction is blocked to reduce the MLL rearrangement of genes (*FLT3*, *PBX3*, *MEIS1*, *HOXA10*, and *HOXA9*) and MLL fusion proteins (CDK6, BCL2, and MEF2C) [17] (Figure 2F).

MI-463 is applied in liver cancer, prostate cancer, breast cancer, and other solid tumors. The IC50 concentration of MI-463 in MDA-MB-231 cells is low (13.99 μM). It was found that compared to the control group, the MI-463 treatment significantly inhibited colony formation and promoted the proliferation of apoptotic cells [26]. Menin is a crucial factor for transcription in breast cancer T47D and MCF-7 cell lines, with poor prognosis that regulates glycolytic genes and oxidative phosphorylation (OXPHOS) expression in tumors. Menin positively regulates *OXPHOS* gene expression. *KMT2A* and two other MAP genes (*MED12* and *GATA3*) are negative regulators of OXPHOS gene expression. The disruption of menin–MLL mediated by MI-503 results in the abrogation of OXPHOS-related gene transcription and a loss of cellular capacity to produce ATP. Members of the menin−MLL complex (menin, MLL1, MED12, GATA3, and WAPL) are important in the transcriptional regulation of glycolysis genes. It was also observed that the reduced transcriptional activity of glycolytic genes with MI-503 may lead to reduced glycolytic enzyme levels (Figure 2G). However, compared to OXPHOS, this transient inhibition had less effect on glycolysis and was quickly recovered. Thus, the MI-503 treatment significantly reduced the amount of ATP produced by OXPHOS, but the existing glycolytic enzymes continued to function and were able to compensate for the energy loss [27].

Meanwhile, accumulating evidence has demonstrated that the menin–MLL complex participates in the tumorigenesis of prostate cancer [17,28]; subsequently, several reports studied the effect of menin–MLL inhibitors on prostate cancer (PCa), such as MI-503 [5,28,29]. It has been demonstrated MI-503 not only inhibited the menin–MLL interaction but also repressed AR signaling in vivo and in vitro in PCa. In vitro, MI-503 reduced the AR target expression, including PSA, TMPRSS2, or FKBP5, subsequently causing the inhibition of AR signaling. In vivo, MI-503 greatly inhibited the proliferation of LNCaP-AR xenograft tumor models as well as castrate-resistant xenografts [5]. Moreover, Huang et al. found that MLL1 and JunD can combine with the same pocket in menin based on the crystal structure analysis [30]. JunD is a crucial regulator for prostate cancer cell proliferation and a latent therapeutic target for PCa [41,42,43]. More intriguingly, treatment with MI503 greatly inhibited the growth of prostate cancer cells and reduced the expression of menin and JunD expression in DU145 and PC3 cells. The reduction in JunD subsequently caused the inhibition of the expression of c-MYC, cell proliferation, migration, and invasion in AR-independent PCa cells (Figure 2H).

According to previous studies of patients with hepatocellular carcinoma (HCC), menin is overexpressed and correlated with poor prognosis [6,44]. Moreover, it has been demonstrated that the protein–protein menin–MLL1 interaction is crucial for HCC growth. MI-503 showed in vitro and in vivo antitumor activity in HCC models. Specifically, MI-503 greatly limits the growth and migration of HepG2 cells. After being combined with sorafenib, MI-503 showed a stronger inhibition of the growth of HepG2 cells. The IC50 values of MI-503 in inhibiting HepG2 cells reached 14 nM. Mechanistically, MI-503 blocked the menin–MLL complex’s ability to bind to the *PEG10* promoter, which is also significant for HCC cells in order to grow and migrate, leading to decreased H3K4 methylation in HepG2 and Hep3B cells [6] (Figure 2I).

In the study of Ewing’s sarcoma, it was found that menin was highly expressed in tumors and acted as an oncogenic factor. The inhibition of menin–MLL1 leads to the extensive and early reprogramming of cell metabolism. The serine biosynthesis pathway (SSP) was influenced the most by MI-503. In Ewing’s sarcoma, high baseline expressions were found in SSP genes and proteins (PHGDH, PSAT1, and PSPH), and metabolic flux was observed via the SSP. Treatment with MI-503 resulted in a decrease in H3K4me3 enrichment on the *PHGDH* promoter, and the de novo synthesis of serine and glycine completely disappeared [31] (Figure 2J). The structures of MI-463 and MI-503 are shown in Table 1.

### 3.5. VTP-50469

It has been reported that, in AML, the most common mutant genes include fms-related tyrosine kinase 3 (FLT3) and nuclear pigment 1 (NPM1). The mutation of FLT3 genes may appear in every AML subgroup, especially NPM1mut AML (60%) and MLL-r leukemia (10%) [45,46]. The IC50 values of VTP-50469 in inhibiting the leukemia cell reach 3 nM. The VTP-50469 targets the expression of FLT3 by inhibiting leukemia transcription factor *MEIS1* and promotes the loss of phosphorylated FLT3 mediated by FLT3 inhibitors, thus producing a synergistic antileukemia impact on NPM1mut or MLL-r leukemia [32] (Figure 2K). VTP-50469 and MI-503 are also used in the treatment of Ewing’s sarcoma [31] (Figure 2J).

### 3.6. BAY-155

The inhibitory effects of BAY-155 on the proliferation of MOLM-13- and MV-4-11-rearranged AML cell lines were 6.3 times and 2.8 times higher than that of MI-503. The IC50 values of BAY-155 in inhibiting the leukemia cell reach 8 nM. BAY-155 downregulated the MEIS1 and up-regulated the differentially expressed MNDA and CD11b in MLL rearranged MOLM-13 and MV-4-11 cells. Overall, compared with MI-503, the improvement in BAY-155 urges researchers to leverage BAY-155 as an inhibitor in order to broadly explore the clinical prospect of menin–MLL1 interactions [33].

### 3.7. KO-539

Treatment with the menin–MLL1 inhibitor KO-539 inhibited the proliferation of OCI-AML3 (expressing mtNPM1), MOLM13 (expressing MLL1-AF9), and MV4-11 cells; induced the expression of bone marrow differentiation marker CD11b; and enhanced the morphological characteristics of cell differentiation. Treatment with KO-539 decreased the expression of BCL2, CDK6, FLT3, MYC, PBX3, and MEIS1 and induced the mRNA and protein levels of CD11b (Figure 2L). Via the ubiquitin–proteasome pathway, the degradation of menin proteins was triggered by the influence of KO-539. It was found that the integrated therapy of KO-539 with BET, CDK6-6, or BCL2-2 inhibitors could induce in vitro synergies that are lethally induced by MLL1-r or mtNPM1 on AML cells [34,35].

### 3.8. MI-1481

MI-1481 inhibited menin interactions with MLL1, achieving an IC50 value of 3.6 nM and about a 10-fold increase in inhibiting MI-503 and MI-463 [17,25]. The proliferation of murine bone marrow cells with MLL-AF9 fusion transformation decreased significantly. MI-1481 also showed human MLL cell growth inhibition in MOLM13 (containing ML-AF9) and MV-4-11 (containing MLL-AF4 fusion protein). In K562 and U937 human leukemia cell lines without *MLL* translocation, no significant inhibition was found in the proliferation of cells. MI-1481-treated human MLL leukemia cell MV-4-11 was employed to detect the influence of the expression levels of *DLX2*, *MEIS1*, and *HOXA9*, which are all MLL fusion target genes. The results show that significant inhibition was observed at the expression level of the MLL fusion target gene (Figure 2F). In addition, after MI-1481 treated MV-4-11 cells, the differentiation marker ITGAM was significantly upregulated, which supported cell reprogramming from primitive cells to more mature blood cells. In conclusion, MI-1481 shows significant targeting activity in both advanced MLL leukemia mouse models and human leukemia cells, which indicates the clinical prospect of this complex [36].

### 3.9. MI-3454

MI-3454 has high inhibitory activity, achieving an IC50 value of 0.51 nM—which can block the interaction between menin and the entire MLL binding fragment—nearly 60-fold more than MI-503. MI-3454 showed significant activity and targeting in MLL leukemia cells. According to existing research, MI-3454 significantly decreased the activity of leukemia cells containing different MLL fusion proteins, including MLL-ENL, MLL-AF4, and MLL-AF9, and the IC50 value was between 7 and 27 nM. It is indicated that among the currently reported menin–MLL1 inhibitors, MI-3454 has the optimum activity in MLL leukemia cells [14,17,20,25,36,47]. On the contrary, the proliferation of MLL1 translocated leukemia cells was not influenced by MI-3454. It was found that MI-3454 can result in the downregulation of *MEIS1* and *HOXA9* expressions in MV-4-11 and MOLM13 cells (Figure 2M). Upon MI-3454 treatment, the expression of *MNDA* genes promoting cell differentiation was upregulated. In addition, MI-3454 significantly reduced the growth of mouse bone marrow cells transformed with MLL-AF9 oncogene and led to the downregulated expressions of HOXA9 and MEIS1. Finally, MI-3454 can induce leukemia regression in MLL leukemia xenotransplantation models by targeting activity. Moreover, influenced by MI-3454, the cloning potential of AML patients with NPM1 mutation or MLL1 translocation was impaired at varying degrees. Taken together, MI-3454 is expected to be an anticancer therapy for *NPM1*-mutated or MLL1-rearranged leukemia [48].

## 4. Type of Macrocyclic Mimics Peptide Inhibitor

### MCP-1

Two types of small molecule inhibitors, hydroxymethylpiperidine and thiophenacil, were obtained via the high-throughput screening method, while macrocyclic peptide-like inhibitors were obtained via different methods. It was found that menin protein mainly binds to the MBM 1 and MBM 2 peptide of MLL fusion proteins. Currently, all small molecule inhibitors mainly combine with the MBM 1 pocket on the menin protein. Zhou et al. analyzed the intermolecular interaction between menin protein surfaces and the MBM 1 peptide and designed a high-activity macrocyclic peptide inhibitor based on the MBM 1 fragment after obtaining the crystal structure of the menin–MLL protein complex [37]. This small molecule inhibitor is directly based on the menin–MLL1 interaction and the design and modification of peptide MBM 1, which greatly reduces the molecular weight of the inhibitor. The IC50 value of the menin–MLL1 interaction was 18 nM. In addition, the Ki value of menin-MCP-1 binding was 4.7 nM, which was 600-fold higher than that of the non-cyclic peptide compounds. MCP-1 also downregulated the expression of MEIS1 and HOXA9 in MV-4-11 and MOLM13 cells. MCP-1 can thus be used as a suitable drug to limit the interaction between menin and MLL1 [37]. The structure o fMCP-1 is shown in Table 3.

## 5. Conclusions and Perspectives

In the past ten decades, the application of menin/MLL1 small molecule inhibitors in treating MLL-rearranged leukemia and solid tumors has been widely studied. In this paper, the main progress of recent research on menin–MLL1 inhibitors was summarized.

Hydroxymethyl/ammethylpiperidine inhibitors are first-generation menin/MLL small molecule inhibitors, which were obtained via high-throughput screening. Although these types of compounds displayed strong inhibitory activity—in particular, the IC50 value of MIV-6R could reach 50 nmol/L [13]—their poor metabolic stability currently limits the in vivo pharmacodynamic research of these small molecule inhibitors.

Thiophenpyrimidine inhibitors are second-generation menin/MLL1 small molecule inhibitors, and they exhibit good oral bioavailability and strong target binding activity. Remarkably, MI-436 and MI-503 have shown good biological activity [18], which is achieved by the continuous optimization of the original MI-136. In vitro, MI-463 and MI-503 showed precise and selective inhibitory activity on MLL translocation leukemia cells, among which the IC50 value of MI-503 could reach 15 nmol/L. Toxicological evaluations showed that with the continuous administration of MI-436 or MI-503 inhibitors on normal mice for 10 days, there was no hematopoietic function damage, and no obvious tissue and organ toxicity was found in mice [18,19]. Overall, MI-463 and MI-503 are two lead compounds with comparable efficacy and drug-like properties. MI-503 has better oral bioavailability and can be tolerated at higher doses in mice, but at this point, we cannot distinguish which compound might represent a better candidate for further development as targeted therapy for the treatment of patients suffering from MLL leukemia. In order to address this question, pharmacological formulations, pharmacokinetic studies, and long-term toxicity in other species should be investigated in the following studies. More intriguingly, the type of thiophenpyrimidine inhibitors was also revealed to be effective in inhibiting the proliferation of solid tumors, including prostate cancer [5,29], liver cancer [6], breast cancer [27], lung cancer [23], and osteosarcoma [50]. However, the molecular mechanism of these menin/MLL1 small molecules inhibiting the proliferation of solid tumor cells is not clear, and it can be roughly understood that they have a synergistic effect with tumor chemotherapy drugs and may participate in the signaling pathway of chemotherapy drugs to inhibit tumors, which comprises speculation and needs to be further proven.

Macrocyclic peptide inhibitors were designed by analyzing the intermolecular interaction between the surface of the menin protein and the MBM1 peptide, which is based on the crystal structure of the menin/MLL1 protein complex. Because this kind of inhibitor is specially modified by the MBM1 peptide, molecular weight is subsequently reduced without affecting the activity of MBM1 peptides. These findings provided a new strategy for the design of menin–MLL small molecule inhibitors. The structural optimization of small molecule inhibitors in this configuration still needs to be continued in order to improve their cell permeability and further reduce their molecular weight before subsequent drug efficacy evaluations can be performed.

Taken together, the development of menin/MLL1 small molecule inhibitors has not made substantial progress due to poor metabolic stability and poor druggability and because the current skeleton structure of inhibitors is relatively monotone. Further investigations are warranted, such as studies starting from a computer-aided drug design method, using the construction of molecular docking and pharmacophore models, and using virtual screening strategies to discover new small molecule inhibitors of menin/MLL1, which may be a promising strategy and a trending research area in the future.

## Figures and Tables

**Figure 1 molecules-28-03026-f001:**
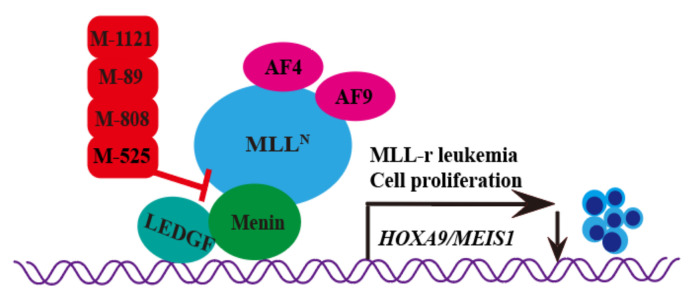
Molecular mechanisms of Hydroxymethyl and aminomethyl piperidine inhibitors in treating leukemia and cancers. Four inhibitors, M-525, M-808, M-89, and M-1121, block the interaction between menin and MLL-AF4, MLL-AF9 fusion protein to reduce the expression of *MEIS1* and *HOXA9* genes and inhibit the proliferation of MLL-r leukemia cells.

**Figure 2 molecules-28-03026-f002:**
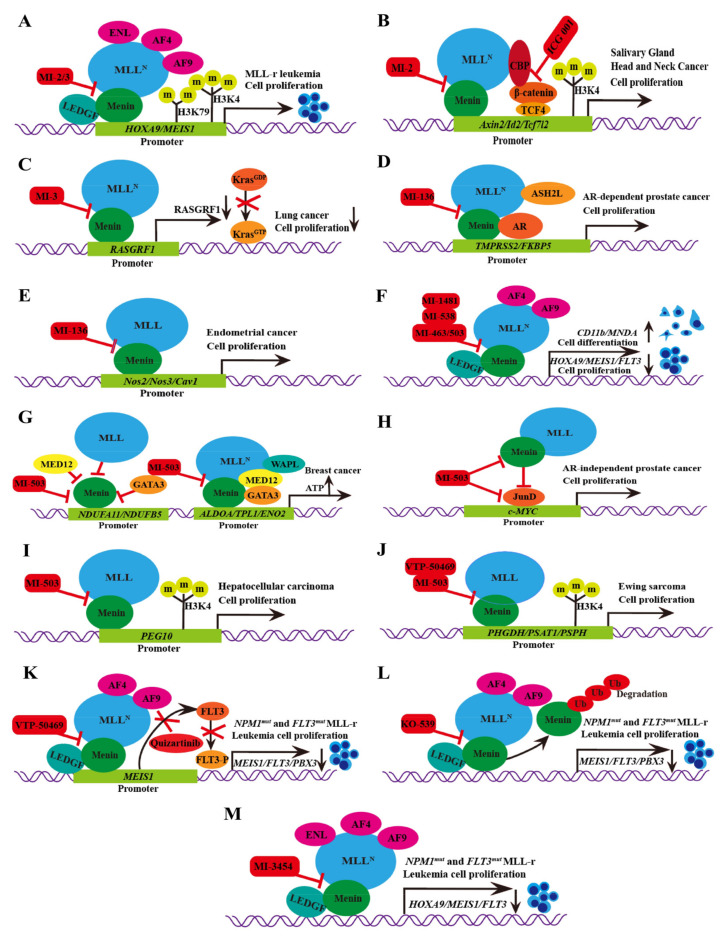
Molecular mechanisms of thiophenpyrimidine inhibitors in treating leukemia and cancers. (**A**) The interaction between menin and MLL-AF4, MLL-AF9, and MLL-ENL fusion proteins is blocked by MI-2 and MI-3 inhibitors to reduce the transcription of *HOXA9* and *MEIS1* genes, which is accompanied by the reduction of H3K4 trimethylation and H3K79 dimethylation and inhibit the MLL-r leukemia cell growth. (**B**) MI-2 blocks the menin−MLL interaction combined with β-catenin inhibitor ICG-001 to reduce the expression of H3K4me3 in the Wnt signaling pathway target genes *Axin2*, *Id2*, *Tcf7l2*, thereby inhibiting the proliferation of salivate gland and head and neck tumor cells. (**C**) MI-3 blocks the menin−MLL interaction and decreases the Rasgrf1 expression. The GTP binding activity of Kras is inhibited, thus limiting the proliferation of *Kras* mutant lung cancer cells. (**D**) MI-136 blocks the menin−MLL interaction to reduce the transcription of AR target genes *TMPRSS2* and *FKBP5*, thus inhibiting the AR-dependent cell growth in prostate cancer. (**E**) MI-136 blocks the menin−MLL interaction to restrict the transcription of HIF signaling pathway target genes *Nos2*, *Nos3*, *Cav1*, thereby inhibiting the proliferation of endometrial cancer cells. (**F**) The interaction between menin and MLL-AF4, MLL-AF9 fusion proteins is blocked by MI-148, MI-538, MI-503, and MI-463, thus reducing the expression of *HOXA9* and *MEIS1* genes and inhibiting the proliferation of MLL-r leukemia cells, respectively. The increased expression of CD11b and MNDA, markers of cell differentiation, promoted the differentiation of leukemia cells into normal cells. (**G**) MI-136 and menin−MLL complex members MLL, MED12, GATA3 synergically inhibited the promoting effect of menin on OXPHOS gene expression. For glycolysis, MI-503 blocks the menin−MLL interaction to reduce the transcriptional activity of glycolysis genes. (**H**) MI-503 blocks the menin−JunD interaction and decreases the expression of menin and JunD. It also inhibits the transcription of JunD target gene c-MYC and the proliferation of AR-independent cells in prostate cancer. (**I**) MI-503 inhibits the interaction between menin and MLL1 to reduce the level of H3K4me3 on the promoter of the *PEG10* gene and limits hepatocellular carcinoma cell growth. (**J**) MI-503 and VTP-50469 inhibitors restrict the menin-MLL1 interaction to reduce the level of H3K4me3 on the promoter of *PHGDH*, *PSAT1,* and *PSPH* genes related to the serine biosynthesis pathway (SSP), subsequently inhibiting the growth of Ewing′s sarcoma cancer cells. (**K**) VTP-50469 blocks the interaction between menin and MLL-AF4, MLL-AF9 fusion proteins to reduce the transcription of the *MEIS1* gene to target the expression of FLT3. Thus, it intensifies the depletion of phosphorylated FLT3 mediated by FLT3 inhibitor and produces synergistic antileukemia impact on NPM1mut or MLL-r leukemia. (**L**) KO-539 blocks the interaction between menin and MLL-AF4 and MLL-AF9 fusion proteins to reduce the expression of *HOXA9* and *MEIS1* genes. Meanwhile, the ubiquitin-proteasome pathway leads to the degradation of menin protein, which has a synergistic antileukemia impact on NPM1mut or MLL-r leukemia. (**M**) MI-3454 blocks the interaction between menin and MLL-AF4, MLL-AF9, and MLL-ENL fusion proteins to reduce the expression of *HOXA9*, *MEIS1,* and *FLT3* genes, which has a synergistic antileukemia impact on NPM1-mut or MLL-r leukemia.

**Table 1 molecules-28-03026-t001:** Summary of Hydroxymethyl and aminomethyl piperidine inhibitors tested against various malignancies.

Classification	Structure	Name	Diseases	Test Model	IC50	Targeting	Reference
Hydroxymethyl and aminomethyl piperidine	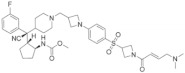	MI-525	Leukemia	In vitro	3 nM	*HOXA9* and *MEIS1*	[14]
	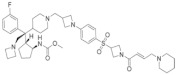	M-808	Leukemia	In vivo In vitro	4 nM	*HOXA9* and *MEIS1*	[15]
	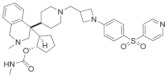	M-89	Leukemia	In vitro	25 nM	*HOXA9* and *MEIS1*	[16,17]
	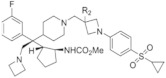	M-1121	Leukemia	In vitro	10.3 nM	*HOXA9* and *MEIS1*	[18]

**Table 2 molecules-28-03026-t002:** Summary of thiophenpyrimidine inhibitors tested against various malignancies.

Classification	Structure	Name	Diseases	Test Model	IC50	Targeting	Reference
Thiophenpyrimidine	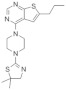	MI-2	Leukemia	In vitro	446 nM	*HOXA9* and *MEIS1*	[20,21]
Head and neck tumors	In vivo	Non determined	Wnt signaling pathway	[22]
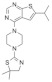	MI-3	Leukemia	In vitro	648 nM	*HOXA9* and *MEIS1*	[20,21]
Lung cancer	In vivoIn vitro	Non determined	Rasgrf 1	[23]
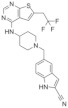	MI-136	Prostate cancer	In vivoIn vitro	5.59 nM	AR signaling pathway	[5]
Endometrial cancer	In vivoIn vitro	4.5 μM	HIF signaling pathway	[24]
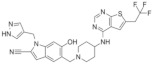	MI-538	Leukemia	In vivoIn vitro	21 nM	*HOXA9* and *MEIS1*	[25]
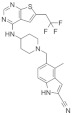	MI-463	Leukemia	In vitro	15.3 nM	*HOXA9* and *MEIS1*	[17]
Breast cancer	In vitro	13.99 μM	Apoptosis	[26]
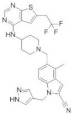	MI-503	Leukemia	In vivoIn vitro	14.7 nM	*HOXA9* and *MEIS1*	[17]
Breast cancer	In vitro	Non determined	Glycolytic and Oxidative phosphorylation	[27]
Prostate cancer	In vivoIn vitro	3.1 μM	AR signaling pathway;menin and JunD	[28,29,30]
Hepatocellular carcinoma	In vivoIn vitro	14 nM	PEG10	[6]
Ewing’s sarcoma	In vitro	3 μM	Serine biosynthesis pathway	[31]
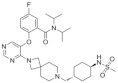	VTP-50469	Leukemia	In vitro	3 nM	*MEIS1* and *FLT3*	[32]
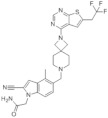	BAY-155	Leukemia	In vitro	8 nM	*MEIS1*	[33]
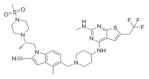	KO-539	Leukemia	In vivoIn vitro	Non determined	*MEIS1*, *FLT3* and *PBX3*	[34,35]
	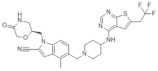	MI-1481	Leukemia	In vitro	3.6 nM	*HOXA9* and *MEIS1*	[36]
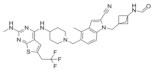	MI-3454	Leukemia	In vivoIn vitro	0.51 nM	*HOXA9, MEIS1* and *FLT3*	[37]

**Table 3 molecules-28-03026-t003:** Summary of Macrocyclic Mimics Peptide inhibitors tested against various malignancies.

Classification	Structure	Name	Diseases	Test Model	IC50	Targeting	Reference
Macrocyclic mimics perptide	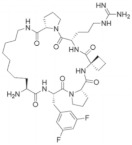	MCP-1	Leukemia	In vitro	18 nM	*HOXA9* and *MEIS1*	[49]

## Data Availability

Not applicable.

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
