# Peer review of "Menin–MLL1 Interaction Small Molecule Inhibitors: A Potential Therapeutic Strategy for Leukemia and Cancers"

_molecules, 2023, doi:10.3390/molecules28073026_

Round 1
Reviewer 1 Report
The review includes a long list of molecules that follow each other in descriptive paragraphs. It would be interesting for the reader if the authors developed the perspectives a little more by formulating comparative opinions between the different inhibitors, highlighting their respective potential and highlighting the most promising. What is missing for the reader is a summary of the state of development of research into MLL inhibitors.
Author Response
Response to Reviewer 1 Comments
The review includes a long list of molecules that follow each other in descriptive paragraphs. It would be interesting for the reader if the authors developed the perspectives a little more by formulating comparative opinions between the different inhibitors, highlighting their respective potential and highlighting the most promising. What is missing for the reader is a summary of the state of development of research into MLL inhibitors.
Response 1: Thank you for your comments on our manuscript. We totally agree with your suggestions and opinions.
We modified the content of Table1 in the new manuscript, including the IC50 values of different inhibitors, inhibitors’ targeting mechanism or pathway of action. We believe that Table 1 could give readers more information about comparative opinions between the different inhibitors now. Meanwhile, we also rewrote the “summary and perspective” section, in order to address your concerns about the summary of MLL inhibitors’ state development, please check and hope that it is now more clearer.
Once again, thank you very much for your constructive comments and suggestions which would help us to improve the quality of the paper.
Kind regards,
LUO Yakun
Email: luoyankun@hrbmu.edu.cn

Reviewer 2 Report
This review summarizes recent advances in the search for small molecule inhibitors of the menin-MLL1 interaction. The molecular mechanisms of these inhibitors and their application prospects in the treatment of AL and cancer were discussed. The article summarizes more comprehensively, the language is also well organized, and the charts can clearly show the content of the review. It is recommended to make some minor revisions.
1. The author should cite more recent relevant research literature.
2. Why does the author say that AL and other tumors are written separately, and whether they can be combined together, so there is no need to talk about the topic separately.
3. It is better to add the mechanism or pathway of action to the entries in Table 1. The in vitro and in vivo tests mentioned by the authors did not reveal more information.
Author Response
Response to Reviewer 2 Comments
This review summarizes recent advances in the search for small molecule inhibitors of the menin-MLL1 interaction. The molecular mechanisms of these inhibitors and their application prospects in the treatment of AL and cancer were discussed. The article summarizes more comprehensively, the language is also well organized, and the charts can clearly show the content of the review. It is recommended to make some minor revisions.
Thank you for your comments on our manuscript. We totally agree with your suggestions.
- The author should cite more recent relevant research literature.
Response : We agree and have updated.
- Why does the author say that AL and other tumors are written separately, and whether they can be combined together, so there is no need to talk about the topic separately.
Response : We thank the reviewer for pointing this out. We apologize that the separated Figure 1 and Figure 2 may cause the misleading for readers. We followed the inhibitors’ type as a writing clues, instead of the different disease, actually. We have revised and combined the Figure 1 and Figure 2 together as the new Figure 1 in the manuscript, in order to address your concerns and hope that it is now clearer.
- It is better to add the mechanism or pathway of action to the entries in Table 1. The in vitro and in vivo tests mentioned by the authors did not reveal more information.
Response : We agree and have revised.
We would like to thank the referee again for taking the time to review our manuscript.
Kind regards,
LUO Yakun
Email: luoyankun@hrbmu.edu.cn

Round 2
Reviewer 1 Report
Thank you for the revised version, I agree that it improves digestion and synthesis of the information for the readers.
Author Response
Dear Review,
We appreciate Editors/Reviewers' warm work earnestly, and thank you very much for your comments and suggestions again.
Best regards,
Luo Yakun